# Effects of synthetic and environmentally friendly fungicides on powdery mildew management and the phyllosphere microbiome of cucumber

Ping-Hu Wu[1], Hao-Xun Chang[1], Yuan-Min Shen[2]*

1 Department of Plant Pathology and Microbiology, National Taiwan University, Taipei City, Taiwan,
2 Master Program for Plant Medicine, National Taiwan University, Taipei City, Taiwan

* shenym@ntu.edu.tw

**Data Availability Statement:** The raw reads were deposited into the NCBI Sequence Read Archive (SRA) database under the accession number: PRJNA874425.

## Abstract

Modern agricultural practices rely on synthetic fungicides to control plant disease, but the application of these fungicides has raised concerns regarding human and environmental health for many years. As a substitute, environmentally friendly fungicides have been increasingly introduced as alternatives to synthetic fungicides. However, the impact of these environmentally friendly fungicides on plant microbiomes has received limited attention. In this study, we used amplicon sequencing to compare the bacterial and fungal microbiomes in the leaves of powdery mildew-infected cucumber after the application of two environmentally friendly fungicides (neutralized phosphorous acid (NPA) and sulfur) and one synthetic fungicide (tebuconazole). The phyllosphere α-diversity of both the bacterial and fungal microbiomes showed no significant differences among the three fungicides. For phyllosphere β-diversity, the bacterial composition exhibited no significant differences among the three fungicides, but fungal composition was altered by the synthetic fungicide tebuconazole. While all three fungicides significantly reduced disease severity and the incidence of powdery mildew, NPA and sulfur had minimal impacts on the phyllosphere fungal microbiome relative to the untreated control. Tebuconazole altered the phyllosphere fungal microbiome by reducing the abundance of fungal OTUs such as Dothideomycetes and Sordariomycetes, which included potentially beneficial endophytic fungi. These results indicated that treatments with the environmentally friendly fungicides NPA and sulfur have fewer impacts on the phyllosphere fungal microbiome while maintaining the same control efficacy as the synthetic fungicide tebuconazole.

## Introduction

Cucumber (*Cucumis sativus*) is one of the most important vegetable crops around the world and in Taiwan. Its cultivation area in Taiwan covers 1,949 hectares, and its production is approximately 47,975 tons annually [1]. In Taiwan, most cucumber is produced in

**Funding:** This research was in part supported by the National Science and Technology Council, Taiwan (grant number 111-2313-B-002 -067). The funders had no role in study design, data collection, and analysis, decision to publish, or preparation of the manuscript.

greenhouses to avoid damage from insect pests, but poor aeration in the greenhouses may accelerate the development of powdery mildew disease. Cucumber powdery mildew is known to be caused by different pathogens, such as *Golovinomyces cucurbitacearum*, *Golovinomyces orontii*, *Podosphaera xanthii* and *Leveillula taurica* [2]. Among them, *P. xanthii* is the most widespread species in tropical and subtropical areas [3], and it is also one of the most important cucumber pathogens in Taiwan [2]. This polycyclic pathogen can infect leaves, shoots, and fruits during all plant growing stages. Gray-white mycelia and conidia can cover the entire plant surfaces of severely infected tissues and reduce photosynthesis, which leads to a reduction in yield and fruit quality [4].

The high inputs of fungicides used to manage cucumber powdery mildew in this continuously harvested crop, which has a short growth period, have aroused public concern. The intensive use of synthetic fungicides increases risks associated with fungicide residues, which can be ingested by humans. It has been shown that the intensive application of synthetic fungicides may stimulate a rapid evolution of fungicide resistance, which may cause the emergence of new races and epidemics in agricultural ecosystems [5,6]. Moreover, resistance could be built up in nontarget environmental fungi. An important example is the overuse of azole fungicides in agricultural environments, which has been linked to antifungal medicine resistance in human pathogens [7,8].

Studies have also demonstrated that synthetic fungicides can have negative effects on biodiversity [9]. For example, the application of tebuconazole, a widely used azole fungicide in agriculture, is reported to decrease soil microbial biomass and activity [10]. Another long-term field study in a vineyard revealed that this group of fungicides has a strong negative effect on biodiversity, especially by affecting the yeast populations on grape berries. Interestingly, in the same study, treatment with sulfur, which is an environmentally friendly fungicide, may preserved the yeast population [11].

At present, the use of environmentally friendly fungicides has been proposed as a substitute for the application of synthetic fungicides, especially for organic farming. The environmentally friendly fungicides that control cucumber powdery mildew include neutralized phosphorous acid (NPA), sulfur, potassium bicarbonate, emulsified oil [12], and other biological control agents. NPA is an elicitor of the plant defense response [13] and has been suggested to effectively control powdery mildew caused by *P. xanthii* [14–16] and *Oidium* sp. [17] on cucumber under greenhouse conditions. It has been increasingly used to control powdery mildew and other crop diseases [15,18]. Sulfur is one of the oldest fungicides and was used to control powdery mildew as early as the nineteenth century. Sulfur can kill spores and mycelia by disrupting respiration and can be used as a preventive or therapeutic fungicide [19,20]. Both NPA and sulfur are generally assumed to present low risks to human health and to the ecosystem because they are natural plant nutrition molecules [21,22]. However, studies on the agricultural and ecological effects of these two environmentally friendly fungicides are still limited.

Because NPA and sulfur are used by foliar spraying, the impact of NPA and sulfur on the phyllosphere microbiome needs to be further addressed. The phyllosphere is colonized by a wide diversity of microorganisms, including bacteria, fungi, and viruses [23]. The composition of the microbiome affects plant physiological processes, such as photosynthesis and disease resistance [24,25]. Recent evidence indicates that the leaf is a reservoir for potential biocontrol microbes, as suggested by the isolation of antagonistic endophytic fungal strains from cucumber leaves against cucumber pathogens [26]. Consequently, research on the phyllosphere microbiome has been conducted to elucidate its important effects on the host and ecosystem.

Extensive studies have shown that the composition of the phyllosphere microbiome can be affected by multiple factors. Although the tissue compartment and host genotype are the most dominant factors [27,28], environmental conditions, stress [23,29], foliar spray [30,31], and

fertilizer application [32] may also alter the microbial abundance and composition. Among these factors, data on foliar spray impacts on the phyllosphere microbiome are limited. In a few studies, synthetic fungicides have shown nontarget effects on phyllosphere yeast [31,33], while their effects on microbial richness and diversity have differed between studies [31,33,34]. In contrast, environmentally friendly fungicides, biological fungicides, and elicitors have had no effect or even serve to increase microbial diversity after they are applied [28,30]. While these results are derived from several different crops, there is still very little understanding of the phyllosphere microbiome of cucumber. Furthermore, comparison of the effects of synthetic and environmentally friendly fungicides on the crop still need to be examined.

Motivated by the growing demand for environmentally-friendly alternatives to synthetic fungicides for sustainable plant disease management, we have investigated the effects of one synthetic fungicide (tebuconazole) and two environmentally friendly fungicides (NPA and sulfur) on the development of cucumber powdery mildew. The dynamics of bacterial and fungal communities in the cucumber phyllosphere have also been analyzed using amplicon sequencing to explore the impacts of the fungicides on the phyllosphere microbiome. This study aimed to (1) determine the relative efficacy of NPA, sulfur, and tebuconazole in controlling cucumber powdery mildew; (2) compare the phyllosphere diversity of bacteria and fungi among different treatments; and (3) identify fungal operational taxonomic units (OTUs) affected by different treatments in the cucumber phyllosphere. To the extent of our knowledge, no other studies have been conducted to evaluate the effect of foliar spray on the cucumber phyllosphere microbiome. This study may shed light on the impacts of different fungicides on the microbial composition in the leaves, providing new insights into the ecological consequences of different fungicides on the disease management of cucumber powdery mildew.

## Materials and methods

### Plant material and experimental design

The experimental trials were conducted from June to August 2020 in a greenhouse at the Taichung District Agricultural Research and Extension Station (Changhua County, Taiwan). The average temperature in the greenhouse was 30.8˚C, with maximum and minimum temperatures of 41.4˚C and 25.5˚C, respectively (S1 Fig). Twelve-day-old seedlings of cucumber cultivar CU87 (Ho Sheng Seed, Tainan, Taiwan) were transplanted into 20 cm x 65 cm pots filled with 7.5 kg peat (Jiffy Products, Norway), and the pots received daily irrigation. The physical and chemical properties of the potted soil were tested at the end of the experiment and are shown in S1 Table. Each treatment contained 15 plants (3 plants x 5 pots). The pots were arranged in a randomized complete block design. The plants were naturally infected by powdery mildew in the greenhouse. Four treatments, consisting of control (water), NPA, sulfur, and tebuconazole, were applied three times after flowering on 21 July, 28 July, and 4 August. Fifty milliliters of liquid solution was spread onto each plant. Preparation of the NPA stock was performed by mixing 100 g of 99% phosphorus acid (Omichi Seiyaku Co., Ltd., Osaka, Japan) and 100 g of 95% potassium hydroxide (Nippon Soda Co., Ltd., Tokyo, Japan) into 200 ml water, and NPA was sprayed at a 250-fold dilution of the stock. Sulfur (80% water dispersible granules, Syngenta Taiwan Co., Taipei, Taiwan) was diluted 1200-fold. Tebuconazole (25.9% emulsion oil in water, Bayer Taiwan Co., Taipei, Taiwan) was diluted to 86.33 ppm (3000-fold). The incidence and severity of powdery mildew disease were evaluated on the day after the last fungicide application. Disease incidence was the percentage of infected leaves to total investigated leaves, and disease severity was estimated according to a six-grade rating scale for each leaf [35]. In brief, the six grades included 0 = 0%, 1 = 0–1%, 2 = 2–5%, 3 = 6–20%, 4 = 21–40%, and 5 = >40% of the leaf area infected. Disease severity (%) = [sum

(grade frequency × score of the grade)]/[(total number of leaves investigated × 5)] × 100. Approximately 36 fully expanded leaves (12 each plant, 3 plants) were inspected to estimate incidence and severity. Each treatment contained five biological replicates. At the end of the experiment, three biological replicates from each treatment were chosen for DNA extraction. Thirty-six leaves collected from each pot (12 leaves on each plant) were pooled into a biological replicate. The leaf samples were then frozen at -20˚C prior to DNA extraction. To verify the potential pesticide residues on cucumber, the fruits were analyzed using a standard testing method for pesticide residues [36] at the Taiwan Agricultural Chemicals and Toxic Substances Research Institute, Taichung, Taiwan. No pesticide residue was detected in the control, NPA and sulfur treatments (the limit of quantification was 0.01 ppm). Tebuconazole (0.23 ppm) was detected on the fruit after tebuconazole treatment.

### DNA extraction and Illumina sequencing

To extract DNA, a total of 3 g of each biological replicate leaf sample was cut from the bottom right of 36 top leaves (83 mg each), immediately frozen in liquid nitrogen and then homogenized in a mortar with a pestle. Two hundred milligrams of homogenized leaf powder was transferred into a 2 ml Eppendorf tube and mixed with 800 µl of buffer XL from a plant DNA isolation kit (Geneaid Biotech Ltd., New Taipei City, Taiwan). Total DNA was extracted using the kit by following the manufacturer's instructions. The integrity of purified DNA was qualified by a Bioanalyzer 2100 instrument (Agilent Technologies, Inc., CA). To reduce interferences from cucumber chloroplasts and internal transcribed spacer (ITS) sequences, amplicon libraries were prepared from a nested PCR protocol according to previously described methods with some modifications [37,38]. In brief, the bacterial 16S ribosomal RNA (rRNA) gene V5-V7 region was amplified using primers 799F and 1391R (Table 1). The 600 bp DNA fragment of the PCR product was purified by gel extraction and reamplified using primers 799F and 1193R (Table 1). The ITS region of the fungus was amplified using primers ITS1F_KYO2 and ITS4 (Table 1). The 600 bp DNA fragments of the PCR products were purified by gel extraction and reamplified using primers ITS3_KYO2 and ITS4 (Table 1). Amplicons were purified with a QIAquick PCR Purification Kit (Qiagen, Hilden, Germany) and then pooled and concentrated with AMPure XP beads (Beckman Coulter, Inc., CA, USA). A paired-end library was constructed using the Celero DNA-Seq Library Preparation Kit (NuGen Diagnostics LLC., CA, USA). The DNA libraries were sequenced using the 300 bp paired-end sequencing of Illumina MiSeq (Illumina, San Diego, CA) at Tri-I Biotech Inc. (New Taipei City, Taiwan). The raw sequence reads can be obtained in the NCBI Sequence Read Archive database under accession number PRJNA874425.

**Table 1. Primers used for amplicon library preparation.**

| Primer name | Sequence (5'-3') | Reference |
| --- | --- | --- |
| **Bacterial 16S** | | |
| 799F | AACMggATTAgATACCCKg | [39] |
| 1391R | gACgggCggTgWgTRCA | [40] |
| 1193R | ACgTCATCCCCACCTTCC | [41] |
| **Fungal ITS** | | |
| ITS1F_KYO2 | TAgAggAAgTAAAAgTCgTAA | [37] |
| ITS4 | TCCTCCgCTTATTgATATgC | [42] |
| ITS3_KYO2 | TAgAggAAgTAAAAgTCgTAA | [37] |

## Data and statistical analysis

All statistical analyses were conducted using R v.4.1.2 [43]. The differences in disease incidence and disease severity among treatments were analyzed using ANOVA, and the normality assumption was checked by the Shapiro test and Q-Q plot. The homoscedasticity assumption was checked using Bartlett's test and the residual plot. The post hoc analysis was conducted by Tukey's HSD test. Spearman's correlation was applied to disease severity and the number of rarefied *Podosphaera* OTU reads.

The quality of raw reads was checked by CLC genomics workbench v10 (QIAGEN, Aarhus, Denmark) and then extracted and merged using mothur V1.35.0 [44] following the MiSeq protocol developed by Kozich et al. [45]. The merged paired-end reads were trimmed by the following parameters: (i) minimum length of 393 (bacteria) and 320 (fungi), (ii) maximum length of 465 (bacteria) and 500 (fungi), (iii) no ambiguities in the sequence length, and (iv) maximum length of 8 homopolymers in the sequence. Chimeras were identified and removed using the UCHIME algorithm [46]. The sequences were then classified using mothur's Bayesian classifier against the SILVA (v138) and UNITE (v8) databases. All the sequences identified as chloroplast, mitochondria, cucumber, and nontarget organisms were removed from bacterial and fungal datasets. The sequences were then clustered into operational taxonomic units (OTUs) based on 97% sequence similarity. OTUs with less than 0.01% relative abundance were removed, and the samples were rarefied to the smallest read number per sample (17,863 for bacteria and 22,535 for fungi) using the R package vegan [47]. The alpha-diversity indices (richness, Shannon index, Chao index, and Simpson index) were calculated by vegan. The differences in alpha diversity between samples were tested by ANOVA. Non-metric multidimensional scaling (NMDS) was performed based on Bray–Curtis distances using vegan and ggplot2 [48]. Permutation multivariate analysis of variance (PERMANOVA) was conducted using the adonis2 function with 999 permutations in vegan using the Bray-Curtis distance. Differentially abundant OTUs between the control and treatment groups were analyzed by the DESeq2 package [49], and only contrasts with adjusted $p$ values below 0.05 were considered to be significant.

## Results

### Effect of NPA, sulfur and tebuconazole on cucumber powdery mildew disease

All treatments significantly ($p < 0.001$) reduced powdery mildew disease incidence (DI) from 36 to 61% (Fig 1A) and disease severity (DS) from 17 to 23% (Fig 1B) compared with the untreated control. Among the treatments, the leaves treated with sulfur showed the lowest DI (33.33 ± 9.42%) and DS (6.67 ± 1.88%). The DI of the NPA treatment was 38.89 ± 20.6%, and the DS was 8.22 ± 4.12%. The leaves treated with tebuconazole showed the highest DI (58.89 ± 18.47%) and DS (13.22 ± 5.66%). The disease severity was significantly correlated with the richness of *Podosphaera* OTUs (R = 0.67, P = 0.017) (Fig 1C).

### Treatment effects on microbial community α-diversity, β-diversity and composition

The final bacterial OTU table contained 266,923 reads and 1,770 OTUs, and the final fungal OTU table contained 346,020 reads and 2,892 OTUs across all 12 samples. After eliminating OTUs with less than 0.01% relative abundance from all samples, 96 OTUs remained in the bacterial table, and 255 OTUs remained in the fungal OTU table.

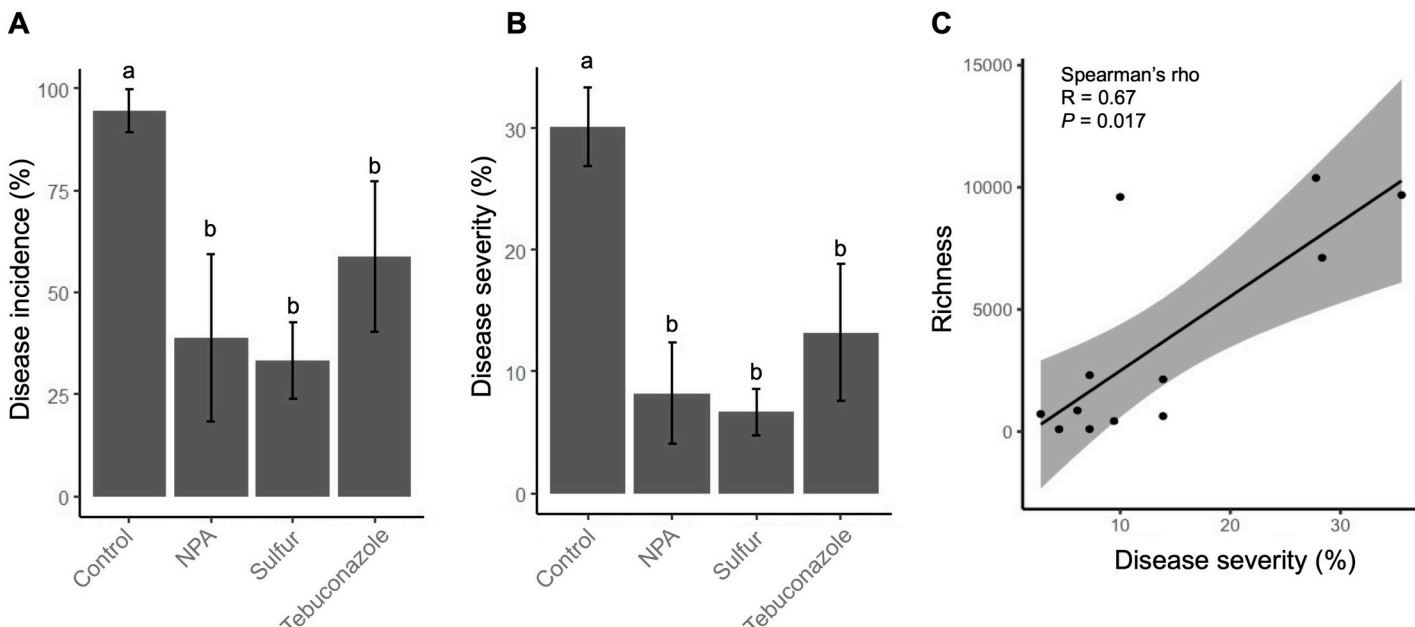

**Fig 1. Effect of treatment on the disease severity and incidence of powdery mildew in cucumber.** (A) Disease incidence, (B) Disease severity 65 days after transplanting and after 3 applications of water (control), NPA, sulfur and tebuconazole. (C), Correlation of *Podosphaera* OTU richness and disease severity. The error bars represent the standard deviations, and the mean separation was conducted using Tukey's HSD test. The same letters indicate no significant difference (α = 0.05).

Five bacterial phyla and 15 fungal phyla were detected in the final OTU table. Within the bacterial phyla, Actinobacteriota and Proteobacteria accounted for more than 93 percent, while in the fungal phyla, Dothideomycetes (48%), Ustilaginomycetes (19%), Leotiomycetes (15%), and Sordariomycetes (12%) were the most dominant taxa. Fig 2 shows the enriched microbial genera that had relative abundances greater than 1%. The most dominant bacterial genera among all the treatments were *Candidatus* Portiera (13% to 31%), *Pseudomonas* (9% to 23%), *Pseudonocardia* (7% to 17%), *Microbacterium* (3 to 11%), *Methylobacterium/Methyloru-brum* (5% to 12%), and *Nocardioides* (2% to 13%). The most dominant fungal genera were *Fer-eydounia* (11% to 23%), *Podosphaera* (1% to 36%), *Nigrospora* (2% to 12%), *Alternaria* (0.9% to 9%), *Curvularia* (3% to 6%), *Cladosporium* (3% to 5%), *Iodophanus* (1% to 3%), *Moeszio-myces* (0.9% to 3%), *Phaeosphaeria* (0.9% to 2%), and *Sympodiomycopsis* (0.1% to 5%).

The alpha diversity of the bacterial and fungal populations among the different treatments is shown in Fig 3. No significant differences were observed in abundance and diversity in both bacterial and fungal communities among all the treatments. In addition, PERMANOVA showed that the fungicide treatment accounted for only 17% variation of the bacterial popula-tion among the samples, although this was statistically non-significant, as evident from the *p*-value of 0.942 (Table 2). However, different treatments significantly affected the fungal commu-nity on cucumber leaves, as the PERMANOVA demonstrated that the treatments accounted for 57% of the variation in the fungal population with a *p*-value of 0.008 (Table 2). The NMDS ordi-nation plot of Bray-Curtis community dissimilarities showed that the NPA and sulfur treat-ments were clustered and distinct from the tebuconazole and control samples (Fig 4B). Since disease severity was correlated with the richness of *Podosphaera* OTUs (Fig 1C) and *Podo-sphaera* OTUs composed 14.6% of the total fungal OTUs, the difference in the fungal commu-nity composition may be mainly leveraged by the *Podosphaera* OTUs. Therefore, to investigate the influence of different fungicide treatments on other nontarget fungal OTUs, the OTU table was reanalyzed after excluding the *Podosphaera* OTUs. The results showed that the different

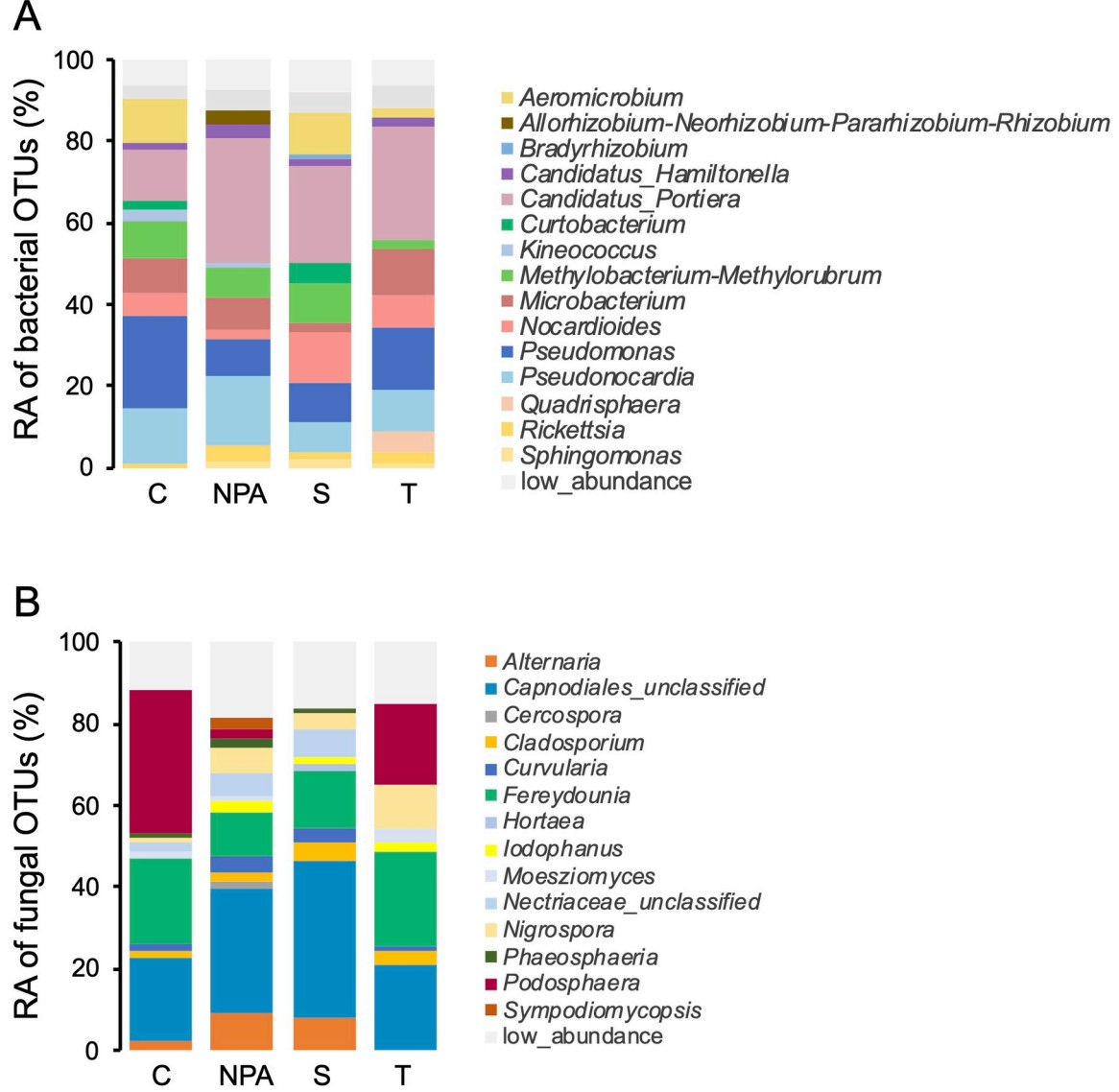

**Fig 2. Bacterial and fungal community structure of different treatments.** Relative abundance (RA) of the most abundant OTUs. (A) bacteria (B) fungi. C: Untreated control, NPA: Neutralized phosphorous acid, S: Sulfur, T: Tebuconazole.

treatments still significantly ($p$ = 0.023) accounted for 45% of the variation in the fungal community after excluding *Podosphaera* (Table 2), but the NMDS ordination plot that excluded the *Podosphaera* OTUs showed that the control, NPA, and sulfur treatments were clustered, while the tebuconazole treatment was distinct from the others (Fig 4C). The results indicated that in addition to *Podosphaera*, which showed reduce abundance after the application of fungicides, other fungi were affected by the treatments, especially by the synthetic fungicide tebuconazole.

## Identification of fungal OTUs affected by different treatments on cucumber

Nineteen fungal OTUs were markedly altered by tebuconazole treatment, whereas 6 and 4 fungal OTUs were altered by NPA and sulfur, respectively, as revealed by the differential DESeq2

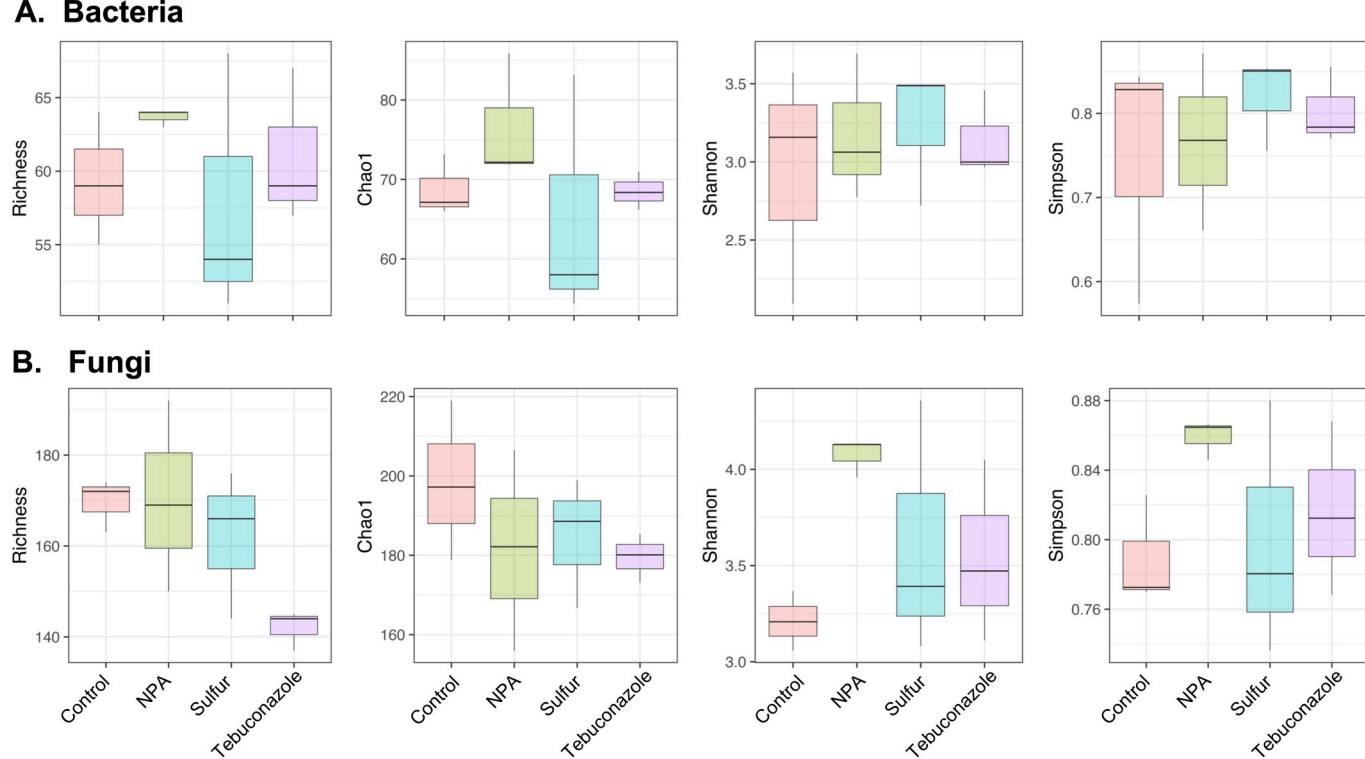

**Fig 3. Alpha diversity of bacterial and fungal populations among different treatments.** The richness (No. of OTUs), Chao1, Shannon index and Simpson index of (A) bacterial (B) fungal populations. NPA: Neutralized phosphorous acid.

analysis compared with the control (Fig 5). Within differential abundance OTUs in tebuconazole versus the untreated control, Sordariomycetes (36.8%) and Dothideomycetes (31.6%) accounted for the largest groups. However, not all Sordariomycetes OTUs decreased in abundance. For example, two *Nigrospora* OTUs increased in abundance after tebuconazole treatment. For NPA and sulfur treatment versus the untreated control, *Podosphaera* OTUs represented the largest genus that decreased in abundance. The yeast-like fungi *Sympodiomycopsis* increased in abundance after both NPA and tebuconazole treatments but decreased in abundance after sulfur treatment (Fig 5).

**Table 2. Permutational multivariate analysis of variance (PERMANOVA) for fungi and bacteria.**

| Target | Factor | Degrees of Freedom | Sum of Squares | $R^2$ | Pseudo-F | *p* |
|---|---|---|---|---|---|---|
| **Bacteria** | Treatment | 3 | 0.345 | 0.173 | 0.558 | 0.942 |
| | Residual | 8 | 1.648 | 0.827 | | |
| | Total | 11 | 1.993 | 1 | | |
| **Fungi** | Treatment | 3 | 0.593 | 0.571 | 3.544 | 0.008 |
| | Residual | 8 | 0.446 | 0.429 | | |
| | Total | 11 | 1.039 | 1 | | |
| **Fungi** **(without *Podosphaera* OTUs)** | Treatment | 3 | 0.322 | 0.453 | 2.211 | 0.023 |
| | Residual | 8 | 0.389 | 0.547 | | |
| | Total | 11 | 0.711 | 1 | | |

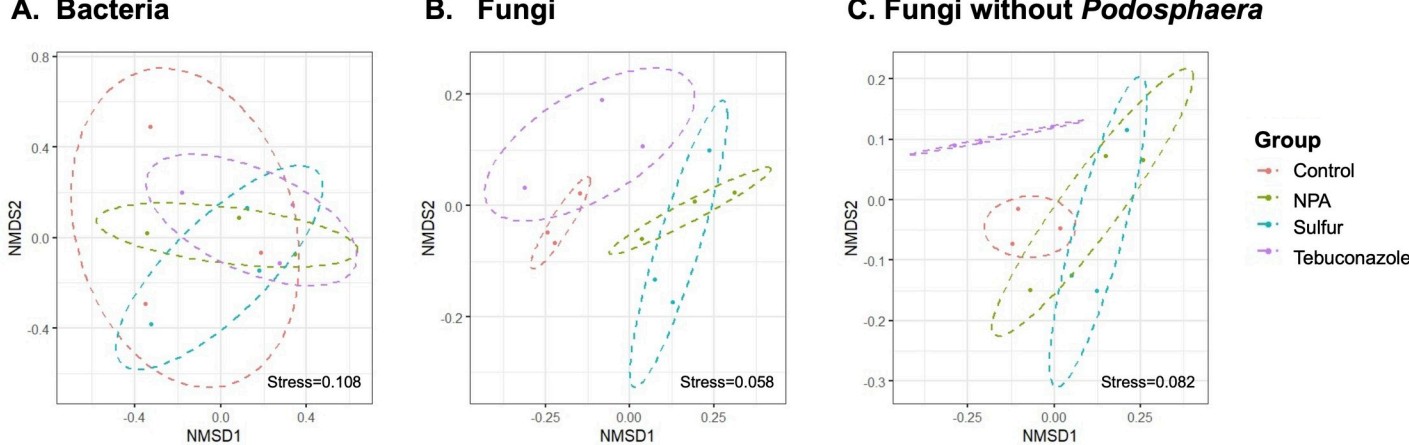

**Fig 4. Effects of fungicides on fungal leaf composition in cucumber.** Non-metric multidimensional scaling (NMDS) ordination plot of Bray-Curtis distance. (A) Bacteria. (B) Fungi. (C) Fungi without *Podosphaera* OTUs. Ellipses represent 99% confidence intervals. NPA: Neutralized phosphorous acid.

## Discussion

In response to the increasing awareness of food safety and global climate change, the requirements for environmentally friendly pesticides have increased in recent years. Organic farming and environmentally friendly fungicide application trends have been increasingly recognized, especially for vegetables such as cucumber. However, limited studies have compared the influence of synthetic and environmentally friendly fungicides on the phyllosphere microbiome. In this study, we demonstrated the changes that occur in the bacterial and fungal microbiome after NPA, sulfur and tebuconazole treatments for managing cucumber powdery mildew. By integrating the disease rating and phyllosphere microbiome analyses, we found that the environmentally friendly fungicides NPA and sulfur had lower levels of impact on the

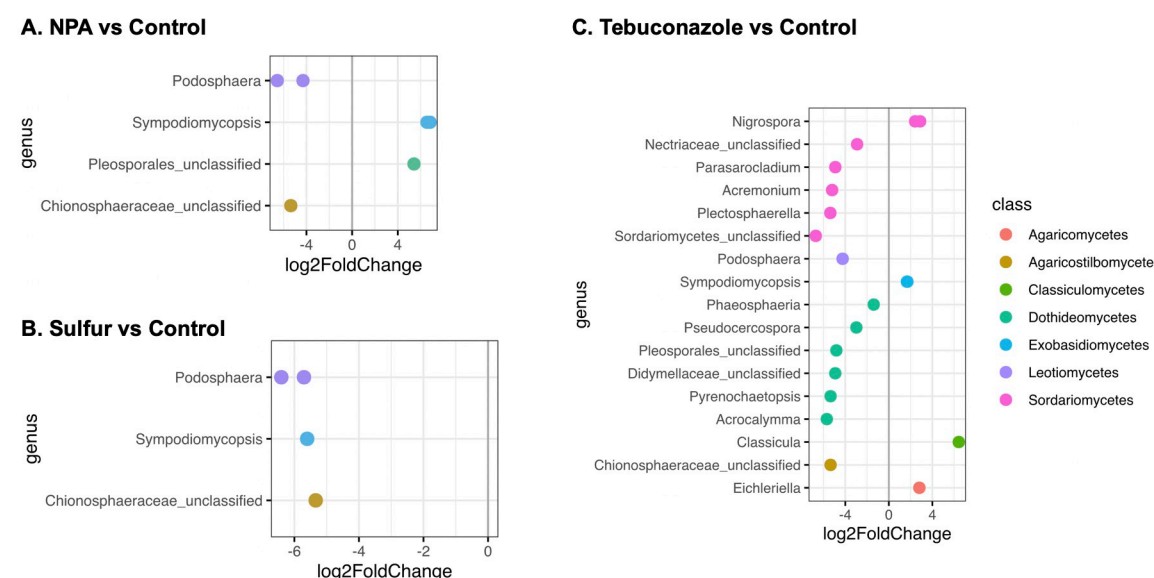

**Fig 5. Fungal genera identified in the differential abundance analysis of different fungicide treatments and untreated controls.** (A) Neutralized phosphorous acid (NPA) vs. control. (B) Sulfur vs. control. (C) Tebuconazole vs. control.

phyllosphere microbial community while achieving the same control efficacy as the synthetic fungicide tebuconazole.

Many environmentally friendly control agents have been tested for cucumber, and the results suggest no significant differences in efficacy relative to synthetic fungicides [12,15,16]. Consistent with these previous studies, we observed that both the plant elicitor NPA and the ancient plant disease control agent sulfur effectively controlled powdery mildew on cucumber and displayed no significant differences from tebuconazole. Therefore, environmentally friendly fungicides can be adopted as alternative control strategies to replace synthetic fungicides.

The leaf microbiome composition of cucumbers was demonstrated under biotic pressure from powdery mildew infection for the first time in this study. The composition of the leaf microbiome can be affected by many factors, but the most important factors may be genotype and the environment. Our experiments represented tropical/subtropical greenhouse production conditions under temperatures as high as 41.4°C (S1 Fig). The results can serve as a reference for comparing with future studies with similar farming conditions. In this study, the dominant bacterial phyla in the cucumber phyllosphere were Actinobacteria and Proteobacteria. This observation was consistent with a previous study on cucumber under different biotic pressures [50]. In both cases, *Pseudomonas*, *Microbacterium*, and *Methylobacterium* were found to be dominant bacterial genera in the cucumber phyllosphere [50]. Notably, *Candidatus* Portiera, *Pseudonocardia* and *Nocardioides* were first reported to be abundant in cucumber leaves in the current study. On the other hand, the most dominant fungal classes in the phyllosphere were Dothideomycetes, Leotiomycetes, Sordariomycetes and Ustilaginomycetes. Dothideomycetes was the most dominant fungal class in cucumber leaves across different studies [50]. Leotiomycetes, which contains the powdery mildew pathogen genera, was the second most abundant fungal class in the untreated control in this study. The powdery mildew pathogen can outcompete the resident leaf-associated fungi and become the dominant group, causing severe disease [29,51]. The most dominant fungal genera were *Capnodiales* (unclassified), *Fereydounia*, *Podosphaera*, *Nigrospora*, *Alternaria*, *Cladosporium*, and *Curvularia*. An OTU belonging to Capnodiales was the largest fungal group found in this study. According to the BLAST result of the representative sequence, the most likely genus for this OTU was *Cladosporium*. Most reported terrestrial *Cladosporium* species are endophytic and some of them are phytopathogens [52]. According to a previous study, *Cladosporium* is one of the major endophytic fungal genera in cucumber leaves [26]. The second largest fungal group observed on the cucumber leaf samples was *Fereydounia*, which is a yeast-like microbe that generally exists in the environment. The possible role of this fungus in cucumber needs to be further examined. The third largest groups were *Podosphaera* in the untreated control and tebuconazole treatment and *Alternaria* in the NPA and sulfur treatment. The content of *Podosphaera* among different treatments may also reflect the control efficacy. Interestingly, the tebuconazole treatment contained 20% *Podosphaera* OTUs, while the untreated control contained 35.5%, the NPA treatment contained 3%, and the sulfur treatment contained 0.8%. This result implies that NPA and sulfur may have better control efficacy than tebuconazole, although there were no significant differences in disease severity.

In this study, we found that NPA, sulfur and tebuconazole did not alter either fungal or bacterial alpha diversity on cucumber leaves. Many studies have demonstrated that the application of synthetic fungicides decreased the richness of fungi in crop leaves and flowers, but the influence of fungicides on the phyllosphere evenness of the fungal microbiome could be different among studies [31,53,54]. Our results showed that there was a tendency for lower richness and diversity in the fungal population of the tebuconazole treatment compared with the other treatments, but the difference was not statistically significant. In the limited number of studies

that investigated the impact of environmentally friendly fungicides on the phyllosphere microbiome, it was found that the application of lime sulfur [30], plant elicitors such as NPA, acibenzolar-S-methyl, laminarin [28], or biopesticides such as *Piriformospora indica* [30] resulted in positive or no impact on phyllosphere microbiome richness or diversity. Current evidence suggests that higher phyllosphere microbiome diversity seems to be an indicator of plant health maintenance. For example, endophytic bacteria (not in total leaves) were less diverse in the *Arabidopsis* leaves of susceptible genotypes [55]. The cocoa genotype that resisted witches' broom showed a greater diversity of symbiont bacteria in its phylloplane than a susceptible genotype [56]. On pumpkin, the richness and diversity of the fungal community on powdery mildew-infected leaves were significantly lower on severely infected leaves than on lightly infected leaves [29]. Therefore, these results are encouraging for plant health, showing that NPA and sulfur control powdery mildew while maintaining the diversity of the phyllosphere microbiome.

The NMDS plot of fungal OTUs further demonstrated that the four treatments were separated into two clusters; the untreated control was clustered with tebuconazole treatment, and the NPA treatment was clustered with sulfur treatment. However, because the treatment targeted the powdery mildew pathogen *Podosphaera*, the pathogen content was in fact affected by the different treatments. Therefore, the difference we observed may be mainly altered by the composition of *Podosphaera* OTUs. To identify whether other fungal OTUs were also affected by different treatments, we carried out PERMANOVA and conducted the NMDS plot without the *Podosphaera* OTUs. PERMANOVA showed that the structure of the bacterial community was similar, but the fungal communities were significantly different among treatments. The DEseq2 analysis results also supported the NMDS plot observation that more fungal OTUs were altered by tebuconazole treatment than by NPA and sulfur treatments.

Concerns regarding the application of synthetic fungicides have mainly centered around the residual effect on human health and negative impacts on ecosystems. A recent study on soybean and maize focused on the influence of phyllosphere microecosystems and revealed that fungicide application had a nontarget effect on the phyllosphere yeast population, which may be correlated with the effect on the physiology of many phyllosphere prokaryotes that are associated with plant health [31]. However, in this study, we did not find any yeast OTUs that were specifically and negatively affected by tebuconazole treatment. In contrast, two yeast OTUs belonging to Exobasidiomycetes were enriched. A study on a cereal phyllosphere also revealed that a few yeasts belonging to Microbotryomycetes and Tremellomycetes were more abundant in fungicide-treated fields [57], and the different results between studies may be due to different types of fungicides and crop systems. In addition, we also found that Dothideomycetes and Sordariomycetes were the two largest groups affected by tebuconazole, while none of the OTUs in these classes were negatively affected by NPA and sulfur treatments. This presents an advantage of using synthetic fungicides: they simultaneously control other potential plant pathogens; however, previous studies also warned that there are many potential biocontrol fungal endophytes in cucurbit leaves, such as *Acremonium*, *Phaeosphaeria* [26] and *Acrocalymma* [26,58], which could be affected by tebuconazole, as shown in this study. Therefore, synthetic fungicides may also eliminate beneficial fungi while controlling pathogens.

Compared with the 19 OTUs altered by tebuconazole treatment, only 6 and 4 fungal OTUs were altered by NPA and sulfur, respectively. The *Podosphaera* OTUs decreased in abundance in both the NPA and sulfur treatments, as expected. However, the *Sympodiomycopsis* OTUs increased in the NPA treatment but decreased in the sulfur treatment. *Sympodiomycopsis* was first isolated from orchid nectar and described by Sugiyama et al. [59]. Currently, four species have been identified under this genus. *Sympodiomycopsis* is a dimorphic Basidiomycetes and mainly stays in the yeast state. The genus is well known for the secretion of glycolipids, which

can inhibit more than 250 fungal species [60]. It has also been reported to be a member of the core microbiome of the rubber tree (*Hevea brasiliensis*) phyllosphere and was hypothesized to be involved in inhibiting pathogenic fungi in the phyllosphere [61]. However, the role of *Sympodiomycopsis* in the cucumber phyllosphere needs to be further investigated.

In this study, we found that NPA suppresses powdery mildew without altering the leaf bacterial and fungal microbiome. This result is consistent with a previous study on grapevine [30] and oak [62]. NPA is a plant elicitor and is used to control plant diseases such as powdery mildew, downy mildew, and *Phytophthora* in various crops [18,20,63,64]. The mechanism(s) by which NPA controls powdery mildew in cucumber remains unclear. Recent studies suggest that NPA suppresses oomycetes mainly by indirectly stimulating host plant defense responses or by stimulating the oomycetes to secrete certain effectors to induce plant defense [65]. A previous study on cucumber downy mildew showed that NPA induced the defense-related genes *thaumatin-like protein* (TLP), *Ribosome-inactivating protein* (RIP) and *Defensin* and may activate downstream defense response and inhibit fungal growth [66]. NPA also increased the content of phytoalexin, phenolic components, flavonoids, anthocyanins and the enzymatic activity of phenylalanine ammonia-lyase (PAL) [66,67]. However, studies have reported that NPA alters the rhizosphere microbiome [68] and enriches antagonistic bacteria that inhibit plant pathogens [69]. The evidence provided in this study shows that NPA does not alter the cucumber phyllosphere microbial community, and this finding supports the specific activity of NPA as a plant defense elicitor.

Although sulfur is thought to be a broad-spectrum fungicide, the concentration of sulfur we applied was effective on *Podosphaera* but not on other fungi in the phyllosphere. This result is in line with previous findings that showed that the application of sulfur has virtually no impact on alpha diversity [30], and the application of sulfur also preserves more yeast on grape berries than the synthetic fungicide penconazole [11]. A possible explanation of the selectivity is that sulfur targets fungal spores or conidia, whereas endophytic fungal mycelia are generally less sensitive to sulfur [20]. In addition, powdery mildew fungi may be more sensitive to sulfur than other fungi [20]. The combined results indicate that sulfur is an environmentally friendly fungicide and is suitable for applications in sustainable agriculture.

## Conclusion

To reduce the negative influence of synthetic fungicides, many effective environmentally friendly fungicides have been proposed as substitutes. In this study, we demonstrate that NPA and sulfur can be alternatives to tebuconazole for controlling powdery mildew in cucumber. Although these fungicides exhibited no effect on the richness and diversity of the leaf microbiome, the environmentally friendly fungicides NPA and sulfur had minimal effects on the phyllosphere fungal and bacterial microbiome compared with the synthetic fungicide tebuconazole. Tebuconazole can reduce other potential plant pathogens, but it also inhibits some potentially beneficial plant endophytes or has a nontarget effect on some fungal species. This study provides new insights into the phyllosphere microbiome and elucidates the impact of different fungicides on disease management and the phyllosphere microbiome during cucumber powdery mildew infections, which can be referenced for the development of sustainable disease control strategies.

## Supporting information

**S1 Fig. Temperature records in the greenhouse.**
(TIF)

**S1 Table. Soil properties of different treatments at the end of the experiments.**
(XLSX)

## Acknowledgments

We are grateful to Cheng-Hsiang Yeh from National Taiwan University and Tung-Ching Huang and Mei-Ling Ho from the Plant Protection Laboratory of Taichung District Agricultural Research and Extension Station (TDARES) for assisting with the experiments. We extend our gratitude to Ya-Wen Kuo at the Soil and Fertilizer Research Laboratory, TDARES, for the analyses of soil properties.

## Author Contributions

**Conceptualization:** Hao-Xun Chang, Yuan-Min Shen.

**Data curation:** Ping-Hu Wu, Hao-Xun Chang.

**Formal analysis:** Ping-Hu Wu, Hao-Xun Chang, Yuan-Min Shen.

**Funding acquisition:** Yuan-Min Shen.

**Investigation:** Ping-Hu Wu, Hao-Xun Chang, Yuan-Min Shen.

**Methodology:** Ping-Hu Wu, Hao-Xun Chang, Yuan-Min Shen.

**Project administration:** Yuan-Min Shen.

**Supervision:** Yuan-Min Shen.

**Validation:** Ping-Hu Wu, Hao-Xun Chang, Yuan-Min Shen.

**Visualization:** Ping-Hu Wu.

**Writing – original draft:** Ping-Hu Wu.

**Writing – review & editing:** Ping-Hu Wu, Hao-Xun Chang, Yuan-Min Shen.

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
