## [Decision Letter · Decision Letter 0]

1 Dec 2022

PONE-D-22-25587Effects of synthetic chemical and environmentally friendly fungicides on powdery mildew management and phyllosphere microbiome of cucumberPLOS ONE

Dear Dr. Shen,

Thank you for submitting your manuscript to PLOS ONE. After careful consideration, we feel that it has merit but does not fully meet PLOS ONE’s publication criteria as it currently stands. Therefore, we invite you to submit a revised version of the manuscript that addresses the points raised during the review process.

We look forward to receiving your revised manuscript.

Kind regards,

Kandasamy Ulaganathan

Academic Editor

PLOS ONE

Journal Requirements:

2. In your Methods section, please provide additional information regarding the permits you obtained for the work. Please ensure you have included the full name of the authority that approved the field site access and, if no permits were required, a brief statement explaining why

"We are grateful to Cheng-Hsiang Yeh from National Taiwan University, Tung-Ching Huang and Mei-Ling Ho form Plant Protection Laboratory of Taichung District Agricultural Research and Extension Station (TDARES) for assisting with the experiments. Thanks are extended to Ya-Wen Kuo, Soil and Fertilizer Research Laboratory, TDARES for the analyses of soil properties."

"This research was supported by the Council of Agriculture, Taiwan. The funders had no role in study design, data collection and analysis, decision to publish, or preparation of the manuscript."

Additional Editor Comments (if provided):

Carefully read all queries raised and address them as desired by the reviewers

Reviewers' comments:

Reviewer's Responses to Questions

**Comments to the Author**

1. Is the manuscript technically sound, and do the data support the conclusions?

Reviewer #1: Yes

Reviewer #2: Yes

2. Has the statistical analysis been performed appropriately and rigorously? 

Reviewer #1: Yes

Reviewer #2: Yes

3. Have the authors made all data underlying the findings in their manuscript fully available?

Reviewer #1: Yes

Reviewer #2: Yes

4. Is the manuscript presented in an intelligible fashion and written in standard English?

Reviewer #1: Yes

Reviewer #2: Yes

5. Review Comments to the Author

Reviewer #1: The paper overlooked previous data on phosphates as inducer of resistance against powdery mildew as followsReuveni, M.; Agapov, V.; Reuveni, R. Suppression of cucumber powdery mildew (Sphaerotheca fuliginea) by foliar sprays of

phosphate and potassium salts. Plant Pathol. 1995, 44, 31–39.

19. Reuveni, M.; Reuveni, R. Efficacy of foliar sprays of phosphates in controlling powdery mildews in field-grown nectarine, mango

trees and grapevines. Crop Prot. 1995, 14, 311–314.

20. Reuveni, M.; Agapov, V.; Reuveni, R. A foliar spray of micronutrient solutions induces local and systemic protection against

powdery mildew (Sphaerotheca fuliginia) in cucumber plants. Eur. J. Plant Pathol. 1997, 103, 581–588

Reuveni, R.; Reuveni, M. Foliar-fertilizer therapy—A concept in integrated pest management. Crop Prot. 1998, 17, 111–118.

Reviewer #2: 1. Extracted DNA of cucumber leaves may contain the genomic sequences of both phyllosphere microbiome and endophytes (especially bacteria and fungi), then how do you distinguish the difference between phyllosphere and endophytic species through your analysis?

(In line 344 authors have mentioned about Cladosporium as one of the major endophytic fungi in cucumber leaves). All the genera identified in this study are phyllosphere residents or even contain endophytes? Kindly, shed some light on this.

2. If endophytes are present then what is the impact of these fungicides on the endophytic diversity present in cucumber leaves as that of phyllosphere microbiome?

3. Species level identification would enhance this study in understanding role of different bacterial species and fungal species in disease management in cucumber.

4. Sulfur is best used by applying it in advance of any outbreak of powdery mildew to protect against the disease, how your study has proven it to be more effective and safe?

6. PLOS authors have the option to publish the peer review history of their article (what does this mean?). If published, this will include your full peer review and any attached files.

Reviewer #1: No

Reviewer #2: **Yes: **Latha Battu

---

## [Author Response · Author response to Decision Letter 0]

29 Dec 2022

Response to Reviewer I

Reviewer #1: The paper overlooked previous data on phosphates as inducer of resistance against powdery mildew as follows

Reuveni, M.; Agapov, V.; Reuveni, R. Suppression of cucumber powdery mildew (Sphaerotheca fuliginea) by foliar sprays of phosphate and potassium salts. Plant Pathol. 1995, 44, 31–39.

19. Reuveni, M.; Reuveni, R. Efficacy of foliar sprays of phosphates in controlling powdery mildews in field-grown nectarine, mangotrees and grapevines. Crop Prot. 1995, 14, 311–314.

20. Reuveni, M.; Agapov, V.; Reuveni, R. A foliar spray of micronutrient solutions induces local and systemic protection against powdery mildew (Sphaerotheca fuliginia) in cucumber plants. Eur. J. Plant Pathol. 1997, 103, 581–588

Reuveni, R.; Reuveni, M. Foliar-fertilizer therapy—A concept in integrated pest management. Crop Prot. 1998, 17, 111–118.

Response: Thank you for your suggestion. We have reviewed these articles and cited them in the manuscript. [Reference 14,15,16,18]

Response to Reviewer II

Reviewer #2: 1. Extracted DNA of cucumber leaves may contain the genomic sequences of both phyllosphere microbiome and endophytes (especially bacteria and fungi), then how do you distinguish the difference between phyllosphere and endophytic species through your analysis?

(In line 344 authors have mentioned about Cladosporium as one of the major endophytic fungi in cucumber leaves). All the genera identified in this study are phyllosphere residents or even contain endophytes? Kindly, shed some light on this.

Response: Thank you very much for the question. The method we used in this study cannot distinguish between leaf epiphyte and endophyte, so we use the term “phyllosphere microbiome” to indicate the microbiome of the whole leaves. (Bashir et al. 2022. https://www.sciencedirect.com/science/article/pii/S0944501321001944). Due to the restriction of the experiment method, we can only speculate whether the microorganism is endophytic or epiphytic according to previous reference. The description of Cladosporium is according to a previous survey of endophyte in cucumber leaves (Huang et al. 2020). 

Thank you for your suggestion; we noticed that the description was too arbitrary. We have rephrased the sentence. [line 358-360]

2. If endophytes are present then what is the impact of these fungicides on the endophytic diversity present in cucumber leaves as that of phyllosphere microbiome?

Response: Thank you very much for the question. Follow by question 1, due to the restriction of the method, we cannot distinguish the impact of the fungicides on epiphyte and endophyte. However, since we found no difference in phyllosphere bacterial population among the three fungicides treatment, we can speculate that the fungicide did not affect epiphytic and endophytic bacteria. For the fungal population, NPA and sulfur treatment may have little effect on endophytic fungus because only a few OTUs were altered after the treatments. Tebuconazole may affect more endophytic fungus because the fungicide can be uptake by plant tissue, and some previously reported endophytic genera were found to be altered after the treatment.

3. Species level identification would enhance this study in understanding role of different bacterial species and fungal species in disease management in cucumber.

Response: Thank you for your suggestion. We have considered extending the analysis to the species level. However, since the primer we used only amplified the V3-V4 region for bacteria and partial ITS for fungus, the information is insufficient to classify these organisms to species level (especially for fungus). Therefore, we remained our analysis and discussion only at the genus level. In the future study, we will use 3rd generation 16S and ITS full-length sequencing to improve the resolution of taxonomy classification.

4. Sulfur is best used by applying it in advance of any outbreak of powdery mildew to protect against the disease, how your study has proven it to be more effective and safe?

Response: Thank you for your question. In this experiment, we applied the fungicides three times after flowering [line 133]. The disease severity was low before the first spraying, so the fungicides were applied in advance of any outbreak of powdery mildew. We have investigated the disease incidence and severity of the three treatments in the manuscript to demonstrate the efficacy of different treatments (Fig 1). The result showed that both indexes of the sulfur treatment were significantly lower than the control but no statistical difference with the NPA and tebuconazole treatment. Therefore, the result can not prove that sulfur was more effective than the other two fungicides. In the manuscript, we stated that the environmentally friendly fungicides "maintained the same effect as the tebuconazole treatment". 

For the safety of sulfur: sulfur is an element that exists in plant tissue and can be used in organic agriculture, so it is seen as a safety fungicide [line 86-88]. In addition, we have done the pesticide residue test [line 153-158], and no fungicide was detected in the sulfur treatment. According to these results, sulfur maintains the same effect as the synthetic fungicide but is safer, so it can replace the synthetic fungicide to control powdery mildew on cucumber.

Additional editing:

To further improve the clarity of the writing, the manuscript has also been reviewed by professional native English-speaking editors. The revision has been made accordingly and the certificate is provided by American Journal Experts under verification code 5DC7-46F6-5C75-8096-8FBP.

---

## [Decision Letter · Decision Letter 1]

1 Feb 2023

PONE-D-22-25587R1Effects of synthetic and environmentally friendly fungicides on powdery mildew management and the phyllosphere microbiome of cucumberPLOS ONE

Dear Dr. Shen,

Thank you for submitting your manuscript to PLOS ONE. After careful consideration, we feel that it has merit but does not fully meet PLOS ONE’s publication criteria as it currently stands. Therefore, we invite you to submit a revised version of the manuscript that addresses the points raised during the review process.

We look forward to receiving your revised manuscript.

Kind regards,

Kandasamy Ulaganathan

Academic Editor

PLOS ONE

Journal Requirements:

Reviewers' comments:

Reviewer's Responses to Questions

**Comments to the Author**

1. If the authors have adequately addressed your comments raised in a previous round of review and you feel that this manuscript is now acceptable for publication, you may indicate that here to bypass the “Comments to the Author” section, enter your conflict of interest statement in the “Confidential to Editor” section, and submit your "Accept" recommendation.

Reviewer #3: (No Response)

2. Is the manuscript technically sound, and do the data support the conclusions?

Reviewer #3: Yes

3. Has the statistical analysis been performed appropriately and rigorously? 

Reviewer #3: Yes

4. Have the authors made all data underlying the findings in their manuscript fully available?

Reviewer #3: Yes

5. Is the manuscript presented in an intelligible fashion and written in standard English?

Reviewer #3: Yes

6. Review Comments to the Author

Reviewer #3: Wu et al. work on “Effects of synthetic and environmentally friendly fungicides on powdery mildew management and the phyllosphere microbiome of cucumber” looks quite interesting. The work and the context are appreciable, and the results deserve publication. However, there are certain points which need to be addressed before considering it for the publication.

1. In the introduction clearly state the importance of the topic and recent examples from the literature. And in the last paragraph of the Introduction, please state your motivation, goals and objectives, the novelty of this research, and potential contribution to the literature.

2. Figures are poorly prepared. The quality of the figure is low and not presentable. The resolution of the figures should be improved to maintain at least a minimum quality of 600 dpi.

3. In Page 10, Table 2 displayed alpha diversity among different treatments. Representing alpha diversity of bacterial and fungal populations among different treatments as an R-plot would be far better to understand the difference.

4. In Page 9, on line 264, in the description about PERMANOVA results, the authors stated that the 264 (R2 = 0.17, P = 0.942). A clear description about these values should be included. For instance, R2 value says that the fungicide treatment accounts for only 17 % variation among the samples which is statistically not significant, as evident from the p-value 0.942. Moreover, “P” should be represented as “p”.

5. Further analyses such as identification of core microbiome from the present data, correlation analysis based network models can greatly improve the understanding and add key findings to the current conclusions.

7. PLOS authors have the option to publish the peer review history of their article (what does this mean?). If published, this will include your full peer review and any attached files.

Reviewer #3: **Yes: **BURRAGONI SRAVANTHI GOUD

---

## [Author Response · Author response to Decision Letter 1]

8 Feb 2023

Reviewer #3: Wu et al. work on “Effects of synthetic and environmentally friendly fungicides on powdery mildew management and the phyllosphere microbiome of cucumber” looks quite interesting. The work and the context are appreciable, and the results deserve publication. However, there are certain points which need to be addressed before considering it for the publication.

1. In the introduction clearly state the importance of the topic and recent examples from the literature. And in the last paragraph of the Introduction, please state your motivation, goals and objectives, the novelty of this research, and potential contribution to the literature.

Response: Thank you for the suggestion. The paragraph was revised as follows:

Page 5, lines 114-129

“Motivated by the growing demand for environmentally-friendly alternatives to synthetic fungicides for sustainable plant disease management, we have investigated the effects of one synthetic fungicide (tebuconazole) and two environmentally friendly fungicides (NPA and sulfur) on the development of cucumber powdery mildew. The dynamics of bacterial and fungal communities in the cucumber phyllosphere have also been analyzed using amplicon sequencing to explore the impacts of the fungicides on the phyllosphere microbiome. This study aimed to (1) determine the relative efficacy of NPA, sulfur, and tebuconazole in controlling cucumber powdery mildew; (2) compare the phyllosphere diversity of bacteria and fungi among different treatments; and (3) identify fungal operational taxonomic units (OTUs) affected by different treatments in the cucumber phyllosphere. To the extent of our knowledge, no other studies have been conducted to evaluate the effect of foliar spray on the cucumber phyllosphere microbiome. This study may shed light on the impacts of different fungicides on the microbial composition in the leaves, providing new insights into the ecological consequences of different fungicides on the disease management of cucumber powdery mildew.”

2. Figures are poorly prepared. The quality of the figure is low and not presentable. The resolution of the figures should be improved to maintain at least a minimum quality of 600 dpi.

Response: Thank you for the suggestion. All the figures were refined to 600 dpi and reprocessed by “PACE” to meet the PLOS ONE requirements as recommended in the journal’s submission guidelines.

3. In Page 10, Table 2 displayed alpha diversity among different treatments. Representing alpha diversity of bacterial and fungal populations among different treatments as an R-plot would be far better to understand the difference.

Response: Thank you very much for the suggestion. The information on alpha diversity in Table 2 has been represented as box plots in Fig 3.

4. In Page 9, on line 264, in the description about PERMANOVA results, the authors stated that the 264 (R2 = 0.17, P = 0.942). A clear description about these values should be included. For instance, R2 value says that the fungicide treatment accounts for only 17 % variation among the samples which is statistically not significant, as evident from the p-value 0.942. Moreover, “P” should be represented as “p”.

Response: Thank you for the suggestion. The description was revised as follows:

Page 10, lines 272-278

“In addition, PERMANOVA showed that the fungicide treatment accounted for only 17 % variation of the bacterial population among the samples, although this was statistically non-significant, as evident from the p-value of 0.942 (Table 2). However, different treatments significantly affected the fungal community on cucumber leaves, as the PERMANOVA demonstrated that the treatments accounted for 57% of the variation in the fungal population with a p-value of 0.008 (Table 2).”

Lines 285-287

“The results showed that the different treatments still significantly (p = 0.023) accounted for 45% of the variation in the fungal community after excluding Podosphaera…”

5. Further analyses such as identification of core microbiome from the present data, correlation analysis based network models can greatly improve the understanding and add key findings to the current conclusions.

Response: Thank you for the suggestion. Because of the limited number of replicates in each treatment, Spearman’s correlation could not be properly estimated in the network analysis. Although other analyses have been considered, the results were beyond the scope of this article. To support the conclusions of this study, figures and tables that were relevant to the main findings have been included in this article and presented according to previous reviewer advice.

---

## [Decision Letter · Decision Letter 2]

23 Feb 2023

Effects of synthetic and environmentally friendly fungicides on powdery mildew management and the phyllosphere microbiome of cucumber

PONE-D-22-25587R2

Dear Dr. Shen,

We’re pleased to inform you that your manuscript has been judged scientifically suitable for publication and will be formally accepted for publication once it meets all outstanding technical requirements.

Kind regards,

Kandasamy Ulaganathan

Academic Editor

PLOS ONE

Additional Editor Comments (optional):

Reviewers' comments:

Reviewer's Responses to Questions

**Comments to the Author**

1. If the authors have adequately addressed your comments raised in a previous round of review and you feel that this manuscript is now acceptable for publication, you may indicate that here to bypass the “Comments to the Author” section, enter your conflict of interest statement in the “Confidential to Editor” section, and submit your "Accept" recommendation.

Reviewer #3: All comments have been addressed

2. Is the manuscript technically sound, and do the data support the conclusions?

Reviewer #3: Yes

3. Has the statistical analysis been performed appropriately and rigorously? 

Reviewer #3: Yes

4. Have the authors made all data underlying the findings in their manuscript fully available?

Reviewer #3: Yes

5. Is the manuscript presented in an intelligible fashion and written in standard English?

Reviewer #3: Yes

6. Review Comments to the Author

Reviewer #3: All the comments have been addressed. The manuscript can be accepted in the present form and it deserves publication.

7. PLOS authors have the option to publish the peer review history of their article (what does this mean?). If published, this will include your full peer review and any attached files.

Reviewer #3: **Yes: **Burragoni Sravanthi Goud

---

## [Editor Report · Acceptance letter]

27 Feb 2023

PONE-D-22-25587R2 

Effects of synthetic and environmentally friendly fungicides on powdery mildew management and the phyllosphere microbiome of cucumber 

Dear Dr. Shen:

I'm pleased to inform you that your manuscript has been deemed suitable for publication in PLOS ONE. Congratulations! Your manuscript is now with our production department. 

Kind regards, 

on behalf of

Dr. Kandasamy Ulaganathan 

Academic Editor

PLOS ONE